# Pressure-tuning of $\alpha$-RuCl$_3$ towards a quantum spin liquid

Q. Stahl[1], T. Ritschel ●[1] ✉, G. Garbarino ●[2], F. Cova[2], A. Isaeva[3,4], T. Doert ●[3] & J. Geck ●[1,5] ✉

The layered material $\alpha$-RuCl$_3$ is a promising candidate to realize the Kitaev quantum spin liquid (QSL) state. However, at ambient pressure, deviations from the perfect Kitaev geometry prevent the existence of the QSL state at low temperatures. Here we present the discovery of a pressure-induced high-symmetry phase in $\alpha$-RuCl$_3$, which creates close to ideal conditions for the emergence of a QSL. Employing a novel approach based on Bragg and diffuse scattering of synchrotron radiation, we reveal a pressure-induced reorganization of the RuCl$_3$-layers. Most importantly, this reorganization affects the structure of the layers themselves, which acquire a high trigonal symmetry. For this trigonal phase the largest ratio between the Kitaev ($K$) and the Heisenberg exchange ($J$) ever encountered is found: $K/J = 124$. Additionally, we demonstrate that this phase can also be stabilized by a slight biaxial pressure. This not only resolves the conflicting reports of low-temperature structures in the literature, but also facilitates the investigation of the high-symmetry phase and its potential QSL using a range of experimental techniques.

Quantum spin liquids (QSLs) are fascinating states of matter characterized by the absence of static magnetic order, even at absolute zero temperature[1–3]. Instead, QSLs exhibit massive quantum entanglement among spins, leading to intriguing non-trivial topological properties and the emergence of fractionalized quasiparticles. Among the notable examples is the renowned Kitaev QSL, where constituent spins can fractionalize into mobile Majorana fermions coupled to conserved $\mathbb{Z}_2$-fluxes[4]. Remarkably, the application of a magnetic field can even lead to a non-abelian QSL, which holds promise for topological quantum computing[5] and has spurred substantial interest in these systems.

As far as specific materials are concerned, several candidates have been identified, including Na$_2$IrO$_3$, various polytypes of Li$_2$IrO$_3$, H$_3$LiIr$_2$O$_6$, and $\alpha$-RuCl$_3$[2]. Among them, $\alpha$-RuCl$_3$ has proven to be particularly promising, because initial indications of a field-induced QSL[6,7] have gained support from observed signatures of quantized thermal Hall conductance[8].

$\alpha$-RuCl$_3$ is a spin-orbit-assisted Mott insulator with honeycomb layers composed of edge-sharing RuCl$_6$ octahedra[9], see Fig. 1. The low-energy magnetism within the honeycomb layers can be described by $j_{\text{eff}} = 1/2$ pseudospins of Ru $4d^5$, coupled via strongly anisotropic magnetic interactions[10,11]. According to the widely accepted super-exchange model[12], pure Kitaev magnetism is present only in highly symmetric RuCl$_3$-layers, where undistorted RuCl$_6$-octahedra with edge-sharing geometry and 90˚ Ru–Cl–Ru bond angles are realized. Below we will refer to this as ideal Kitaev geometry.

At ambient pressure, however, $\alpha$-RuCl$_3$ adopts a monoclinic crystal structure of $C2/m$ symmetry, with Ru–Cl–Ru bond angles different from 90˚ and a distorted Ru-honeycomb layer. Due to these deviations from the ideal Kitaev geometry, other magnetic couplings, such as the Heisenberg exchange $J$ and the off-diagonal symmetric exchange couplings $\Gamma$ and $\Gamma'$, come into play[13,14]. Consequently, $\alpha$-RuCl$_3$ does not exhibit a Kitaev QSL at low temperatures and zero magnetic fields but instead displays antiferromagnetic zigzag ordering below

[1]Institut für Festkörper- und Materialphysik, Technische Universität Dresden, Dresden, Germany. [2]European Synchrotron Radiation Facility, Grenoble, France. [3]Fakultät für Chemie und Lebensmittelchemie, Technische Universität Dresden, Dresden, Germany. [4]Van der Waals - Zeeman Institute, Institute of Physics, University of Amsterdam, Amsterdam, The Netherlands. [5]Würzburg-Dresden Cluster of Excellence ct.qmat, Technische Universität Dresden, Dresden, Germany. ✉e-mail: tobias.ritschel@tu-dresden.de; jochen.geck@tu-dresden.de

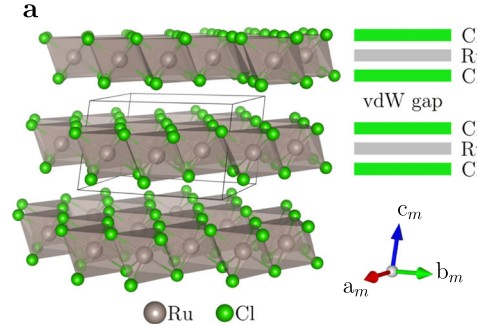

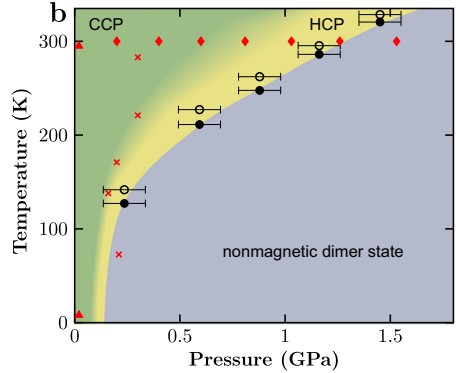

**Fig. 1 | Structure and layer stacking in α-RuCl₃ as functions of temperature and pressure. a** Illustration of the three-dimensional monoclinic *C2/m* structure of α-RuCl₃ at ambient conditions. **b** Pressure-temperature phase diagram of α-RuCl₃. The red triangles and diamonds indicate points where full single-crystal diffraction data sets have been recorded at ambient and at increased pressure, respectively. The red crosses indicate points where overview scans have been collected (further details can be found in the Methods section). The *C2/m* symmetry with cubic close-packed (CCP) Cl is preserved in stress-free samples down to 3 K, as illustrated by the green region. Yellow indicates the region where the phase with hexagonal close-packed (HCP) Cl is stable. With further increasing pressure, a nonmagnetic dimer state is reached. The solid and open black circles, taken from ref. 20, mark the reduction of the magnetic susceptibility upon cooling and warming at constant *p*, respectively. Error bars represent the systematic uncertainty of the pressure measurement.

$T_N = 7$ K[15]. However, while the room-temperature crystal structure is well established, the exact low-temperature structure is still under debate[16–20].

The fact that structural distortions introduce additional magnetic couplings and, hence, are connected to deviations from the pure Kitaev magnetism, prompts the question of whether it is possible to manipulate the lattice of α-RuCl₃ towards the desired ideal geometry. By strategically manipulating the lattice structure, it may, in fact, be possible to stabilize the QSL state – a crucial step in the quest for a reliable material platform for topological quantum computing.

Indeed, the structure of layered van der Waals materials can be effectively modified through various techniques such as pressure and strain engineering or even Moiré engineering[21]. Previous pressure-dependent experiments on α-RuCl₃ have demonstrated that its atomic and magnetic structure is very sensitive to the application of hydrostatic pressure[20,22–25]. In the present study, we take up this approach and extend it based on the phenomenon of polytypism, which is commonly observed in layered transition metal trihalides[26]. Polytypism refers to the fact that the structures of these materials can exhibit different possible stackings of strongly covalent layers, which in turn provides an effective handle to manipulate the magnetic properties of these materials[25,27–29]. A recent Raman study has demonstrated that the application of hydrostatic pressure *p* can alter exactly this stacking in α-RuCl₃[25]. At the same time, pressure-dependent heat capacity measurements have shown that the antiferromagnetic order vanishes at a critical pressure of about 0.7 GPa yielding a new phase of α-RuCl₃[23,25]. Even though the space group P3₁12 has been suggested for this new phase, the precise atomic positions within the covalent Cl-Ru-Cl layers remain unknown. However, this structural information is very important in order to understand the magnetic properties of α-RuCl₃ and is the key result of our study. Remarkably, we will demonstrate in the following sections that the application of *p* drives α-RuCl₃ towards the ideal Kitaev geometry, resulting in a high-symmetry phase that exhibits the largest reported *K/J* ratio to date and essentially ideal $j_{\text{eff}} = 1/2$ states.

## Results

Previous investigations of α-RuCl₃ have demonstrated that a broken-symmetry state characterized by ordered Ru–Ru dimers, known as a valence bond crystal, is stabilized above a critical hydrostatic pressure $p_c(T)$[24,30], see Fig. 1b. In this study, we focus on the detailed examination of the lattice structure within the pressure regime $p < p_c$, beginning with measurements conducted at room temperature, depicted by the red data points in Fig. 1b.

Consistent with previous reports, our diffraction patterns taken at ambient conditions can be indexed by a monoclinic unit cell (space group *C2/m*) with lattice parameters $a_m = 5.9875(6)$ Å, $b_m = 10.3529(3)$ Å, $c_m = 6.0456(6)$ Å and $\beta = 108.777(9)$. However, the crystal studied at room temperature (crystal 2) showed, apart from sharp Bragg reflections, a set of one-dimensional diffuse scattering rods at zero pressure, cf. 04*l*-streak in Fig. 2. These rods are oriented parallel to $c_m^*$, thus revealing a disordered stacking of the RuCl₃-layers along the $c_m$ direction, which can be attributed to stacking faults, characterized by $b_m/3$ shifts between adjacent layers[31].

To characterize the stacking faults and polytypism in α-RuCl₃, we adopt a hexagonal set of basis vectors $a_h$, $b_h$, and $c_h$, where $a_h$ and $b_h$ are parallel to the Cl–Ru–Cl layer, and the $c_h$-axis is perpendicular to it (cf. Fig. 3). Although this representation neglects the small monoclinic distortion present in certain regions of the phase diagram, it allows us to leverage the theoretical framework for X-ray diffraction (XRD) analysis of transition metal trihalides with stacking faults, as developed by Ulrich Müller and Elke Conradi[32].

In Fig. 2, we show the evolution of the x-ray intensity distribution in reciprocal space during the pressure-induced rearrangement of the RuCl₃-layers at room temperature. In this figure and throughout the subsequent analysis, the indexing of reflections is based on the approximate hexagonal cell described in the previous paragraph. To investigate changes in the layer stacking along the $c_h$ direction, we projected the intensities within a slice defined by $-0.02 \leq \Delta h \leq 0.02$ onto the *kl*-plane. The reflections observed in these reciprocal space maps can be classified into three distinct families based on the values of *h* and *k*:

Family 1: $h = 3n$ and $k = 3m$ (with *n*, *m* integers). This family is represented by the 03*l*-streak shown in Fig. 2a–f, where it can be observed that these reflections appear at integer *l* positions and stay sharp along the *l*-direction over the entire pressure range shown.

Family 2: $h - k = 3n$ with *n* integer and *h, k* ≠ 3*n*. These peaks are represented by the -15*l*-streak in Fig. 2g–l and, according to ref. 32, provide insights into the Cl packing. At a pressure of 0.2 GPa, these reflections are sharp and centered at $l = 1/3 + n$, indicating a cubic close packing of Cl, which corresponds precisely to the monoclinic *C2/m* crystal structure of α-RuCl₃ under ambient conditions. However, above 0.2 GPa, a notable broadening of the peaks along *l* is observed, accompanied by a shift of the intensity maximum along the *l*-direction, as depicted in Fig. 2h–j. This signals the loss of long-range order of the

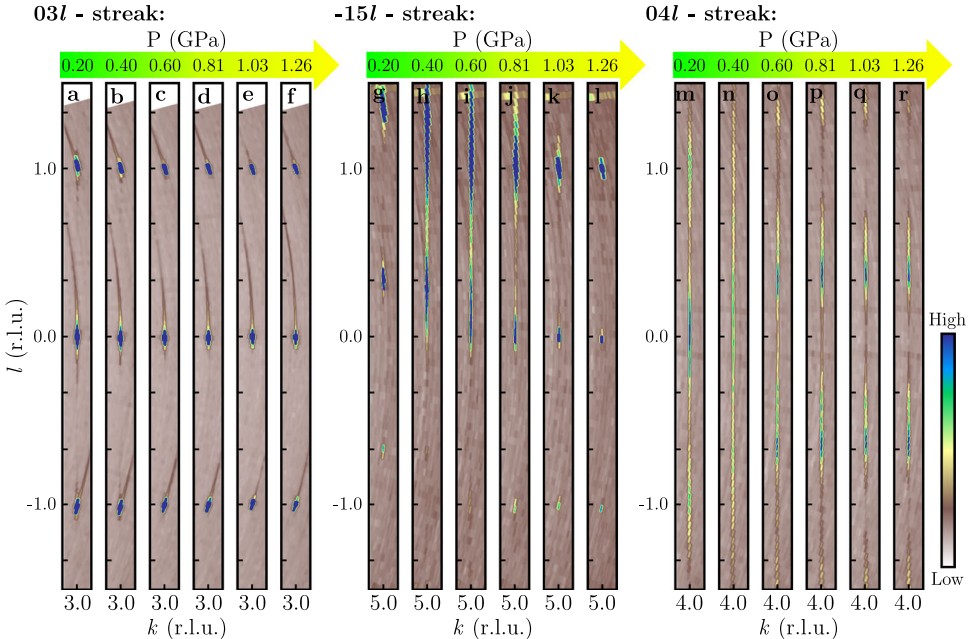

**Fig. 2 | Evolution of the XRD pattern during the pressure-driven phase transition.** Reciprocal space maps parallel to the 03$l$-plane (**a**–**f**), the -15$l$-plane (**g**–**l**) and the 04$l$-plane (**m**–**r**) (integration thickness perpendicular to the plane: $\Delta h = 0.04$) in the pressure range 0.2 GPa to 1.26 GPa at 300 K. The indexation corresponds to the hexagonal setting (cf. Fig. 3). The diffuse stripes along the $l$-direction observable in (**h**–**j**) and (**m**–**r**) are a hallmark of stacking faults. The intensity shift along the $l$-direction in (**g**–**l**) is representative of all reflections with $h - k = 3n$ ($n$ integer and $h, k \neq 3n$) and reveals the rearrangement of the Cl−Ru−Cl sandwich layers from a cubic to a hexagonal closed chlorine packing. While the diffuse stripes along the $l$-direction fulfilling the condition $h - k = 3n \pm 1$ (with $n$ integer) as depicted in (**m**–**r**) signal the disordered stacking of the Ru honeycomb nets over the entire pressure range.

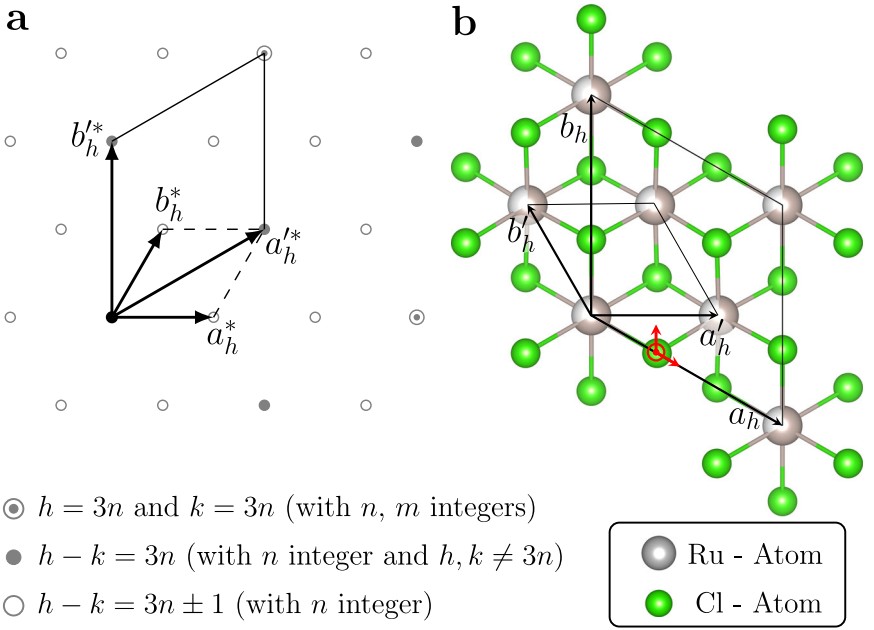

$\circledcirc$ $h = 3n$ and $k = 3n$ (with $n, m$ integers)

$\bullet$ $h - k = 3n$ (with $n$ integer and $h, k \neq 3n$)

$\circ$ $h - k = 3n \pm 1$ (with $n$ integer)

**Fig. 3 | Schematic illustration of the $hk0$-layer and the relation between the real and the average layer structure of $\alpha$-RuCl$_3$ at 1.26 GPa. a** The location of diffuse rods and sharp reflections as a function of pressure leads to three distinguishable families of reflections, indicated by different symbols. The schematic diffraction pattern shows the reciprocal lattice vectors for both the average ($a_h^{\prime *}, b_h^{\prime *}$) and the real ($a_h^*, b_h^*$) layer structure. **b** The average layer structure deduced from the sharp Bragg reflections is marked by the in-plane basis vectors $\mathbf{a'}_h$ and $\mathbf{b'}_h$. The average structure consists of chlorine atoms in a hexagonal-close-packing arrangement in which the Ru atoms occupy all octahedral voids within a layer statistically with a site occupation factor of 2/3. The Cl-displacements $\delta x_{Cl}$ and $\delta y_{Cl}$ from the average position in the real layer structure are indicated by red arrows parallel to $a_h$ and $b_h$, respectively.

Cl-sites in the out-of-plane direction. Notably, the intensity maximum shifts towards integer $l$ values between 0.40 GPa and 0.81 GPa. As the pressure is further increased, the peaks progressively sharpen until they become resolution-limited again at 1.26 GPa. This signifies the emergence of a new long-range ordered state characterized by the reflection condition $l = n$ (with $n$ integer) and peak widths comparable to those of the first family. These data hence reveal a rearrangement of the RuCl$_3$-layers along $c_h$, where the layer stacking with cubic Cl close

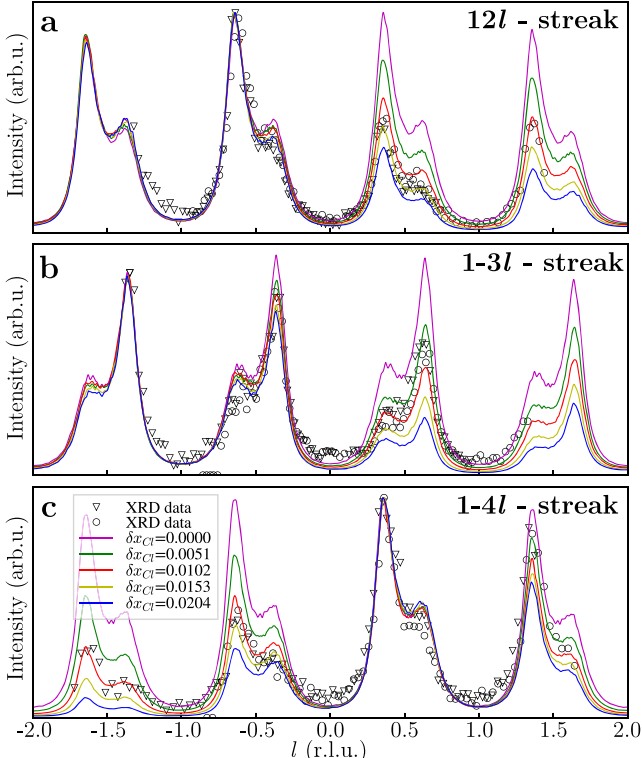

**Fig. 4 | Comparison of calculated diffuse intensity profiles to experimental data taken at 1.26 GPa.** Diffuse intensity profiles along the $12l$ (**a**), $1–3l$ (**b**), and $1–4l$ (**c**) streaks (black circles) and the inverted symmetry-equivalent profiles (black triangles). The intensity profiles were corrected for background intensity as well Lorentz and polarization corrections were applied (see Supplementary Note 2). Solid lines represent simulations of the diffuse intensity where the $Cl$ atom is displaced in 3 pm steps (equals $\delta x_{Cl} = 0.0051$ in fractional coordinates) parallel to $a_h$ from the average position $x_{Cl} = 1/3$ deduced from the sharp reflections. The Miller index $l$ refers to the hexagonal basis vector $c_h$.

packing becomes destabilized with increasing $p$ and eventually transitions to a layer stacking with hexagonal Cl close packing. Notably, this pressure-induced transition between the two long-range ordered Cl lattices is accompanied by huge parallel shifts of the RuCl₃-layers by approximately 2 Å along *ab*.

Family 3: $h − k = 3n \pm 1$ with $n$ integer. This family is represented by the $04l$-streak in Fig. 2m–r, which probes the stacking of the Ru honeycomb layers along $c_h$[32]. The intensity of these reflections is very broad and diffuse along $l$ already at 0.2 GPa, but a maximum at $l = n$ is still discernible. This implies the stacking of the Ru honeycomb layers along $c_h$ is already disordered at 0.2 GPa, which is consistent with the presence of stacking faults in this crystal at ambient pressure (crystal 2). Notwithstanding, the maxima at $l = n$ shows that the sample exhibits the expected $C2/m$-structure at ambient pressure, although with stacking faults.

Despite the disorder in the Ru-honeycomb layer stacking, the stacking of the Cl layers remains fully ordered in all spatial directions at ambient pressure and at $p = 1.26$ GPa. This is due to the fact that for a fixed Cl-order, the Ru-layers, which are sandwiched between the Cl-layers, can still assume different positions. Upon increasing pressure, the intensity distribution initially becomes completely smeared out along $l$, but then, with further increasing pressure, accumulates in broad maxima at $l = n + 1/3$ and $l = n + 2/3$. Further representative intensity distributions for the $3^{rd}$ family of reflections at 1.26 GPa are given in Fig. 4a–c, where the diffuse maxima for different $h − k = 3n \pm 1$ are shown. With increasing $p$ to 1.5 GPa, we observe the transition to the valence bond crystal state characterized by ordered Ru-Ru dimers consistent with previous reports[24,30] (see also Supplementary Note 1).

Taken together, the data presented in Fig. 2 uncover a pressure-induced rearrangement of the RuCl₃-layers, which results in an ordered hexagonal close packing of chlorine atoms but lacks a long-range ordered stacking of the Ru honeycomb layers along $c_h$. To overcome the challenge of determining the structure of the RuCl₃-layers in the presence of structural disorder, we employ a two-step analysis that involves analyzing both the sharp Bragg peaks and the diffuse scattering: The first step is to determine the averaged structure of a single Cl−Ru−Cl layer at room temperature and $p = 1.26$ GPa from the sharp Bragg peaks alone (family 1 and 2), cf. Fig. 3a. However, analyzing the Bragg scattering alone does not provide a complete determination of the structure. Thus, the second step, which involves analysis of the diffuse scattering, is necessary. This enables us to fully determine the structure of the RuCl₃-layers and, at the same time to characterize the stacking faults. Lastly, we compare our experimental findings to density functional theory and observe excellent agreement, thus confirming the accuracy of our analysis of the diffracted intensities.

The Bragg intensity distribution can be modeled in terms of the averaged RuCl₃-layer shown in Fig. 3b, which corresponds to a projection of all atomic sites along the $c_h$-direction onto a single layer. Note that the Ru-sites in this average layer do not form a honeycomb net and that all Ru-sites possess a site occupation factor of 2/3. These layers are then stacked on top of each other along $c'_h$ to form an averaged three-dimensional model structure. Analyzing the positions of the sharp Bragg peaks only, we find that the three-dimensional averaged structure is described by a trigonal cell with lattice parameters $a'_h = b'_h = 3.4080(4)$ Å and $c'_h = 5.562(10)$ Å.

No systematic extinction condition was observed, but the analysis of intensities consistently indicated the trigonal Laue class $\bar{3}m1$. The averaged structure was subsequently solved and refined in the $P\bar{3}m1$ space group, with the detailed information provided in Supplementary Note 1. Due to the small atomic displacements and potential parameter correlations, our crystallographic refinements resulted in averaged atomic positions rather than split positions. From the atomic positions determined in this way, the real structure of the honeycomb RuCl₃-layers can then be reconstructed straightforwardly, assuming the correct stoichiometry: Ru (1a) at (2/3, 1/3, 1/2 + $\delta z_{Ru}$) and Cl (1a) at (1/3 + $\delta x_{Cl}$, $\delta y_{Cl}$, 0.740).

The layer-group symmetry of the above model is $p\bar{3}1m$ for $\delta z_{Ru} = \delta y_{Cl} = 0$. Let us consider possible symmetry reductions due to a finite $\delta z_{Ru}$ or $\delta y_{Cl}$: $\delta z_{Ru} \neq 0$ in combination with the layer disorder discussed above, inevitably results in diffuse streaks along $l$ for all combinations of the Miller Indices $h$ and $k$ at 1.26 GPa, which is at variance with the results presented in Fig. 2 for family 1. From the absence of diffuse streaks for all reflections with $h = 3n$ and $k = 3m$, we therefore conclude $\delta z_{Ru} = 0$. As described previously[33], a finite $\delta y_{Cl} > 0$ would break the symmetry around the Ru-site, which contradicts the condition $\delta z_{Ru} = 0$. The latter, therefore, also implies that $\delta y_{Cl}$ must vanish as well. As a result, we obtain a trigonal cell with lattice parameters $a_h = b_h = 5.9028(4)$ Å and $c_h = 5.562(10)$ Å. The asymmetric unit of this structure contains one Ru-site at (2/3, 1/3, 1/2) and one Cl-site at (1/3 + $\delta x_{Cl}$, 0, 0.740).

To determine the displacement $\delta x_{Cl}$ at $p = 1.26$ GPa, a detailed quantitative analysis of the diffuse scattering along $l$ was conducted. The experimental data was compared to diffuse scattering patterns generated from disordered model structures, which were derived from the previously determined RuCl₃-layers. In order to construct the model structures, stacks of 1000 RuCl₃-layers along the $c_h$-direction were generated using a first-order Markov process, which ensured ordered hexagonal close-packed Cl layers and introduced disordered Ru honeycomb layers. Additional information regarding the construction methodology can be found in Supplementary Notes 2 and 3. The displacement $\delta x_{Cl}$ was then determined by fitting the model structures to the measured intensity profiles. This iterative process

**Table 1 | Structural parameters as determined from the single crystal data and structural optimization calculations as a function of pressure at 300 K**

| | | 0 GPa - $C2/m$ | | 1.26 GPa - $p\bar{3}1m$ | |
|---|---|---|---|---|---|
| | | Experiment | DFT | Experiment | DFT |
| Ru | x | 0 | 0 | 2/3 | 2/3 |
| | y | 0.16651(2) | 0.16633 | 1/3 | 1/3 |
| | z | 1/2 | 1/2 | 1/2 | 1/2 |
| Cl$_1$ | x | 0.22680(13) | 0.22618 | 0.3435(17) | 0.34326 |
| | y | 0 | 0 | 0 | 0.00024 |
| | z | 0.73488(12) | 0.73482 | 0.7400(20) | 0.74149 |
| Cl$_2$ | x | 0.25058(10) | 0.25005 | | |
| | y | 0.17411(4) | 0.17466 | | |
| | z | 0.26761(9) | 0.26788 | | |
| Ru-Ru (Å) | | 3.4477(5) / 3.4570 (4) | 3.4440 / 3.4589 | 3.4080(3) | 3.4080 |
| Ru-Cl-Ru (°) | | 93.62(3) / 94.04(3) | 93.63 / 94.25 | 92.8(4) | 92.73 |

At ambient pressure, the lattice parameter is $a_m = 5.9875(6)$ Å, $b_m = 10.3529(3)$ Å, $c_m = 6.0456(6)$ Å and $\beta$=108.777(9), and the structure is described by the space group $C2/m$. At 1.26 GPa, the lattice parameters are $a_h = b_h = 5.9028(4)$ Å and $c_h = 5.562(10)$ Å, and the structure of a single Cl-Ru-Cl layer is described by the trigonal layer group $p\bar{3}1m$. We note that the fractional coordinate we obtain from structural optimization calculations for the chlorine atom very slightly deviates from the $p\bar{3}1m$ symmetry. Illustrations of the ambient and high-pressure structural models are shown in Fig. 6a–d, respectively.

allowed for the identification of the most suitable value of $\delta x_{Cl}$ that best matched the experimental data.

In Fig. 4a–c, we show representative intensity profiles of reflections belonging to family 3 at 1.26 GPa. As can be observed very nicely in this figure, the calculated diffuse intensity profiles depend very sensitively on $\delta x_{Cl}$, which enables its precise determination via comparison to the experiment. We find the best overall agreement between model and experiment for a Cl-displacement $\delta x_{Cl} = 0.0102$, corresponding to the solid red lines in Fig. 4a–c.

The structural parameters determined from XRD for both the monoclinic phase at ambient pressure and the high-symmetry phase at $p = 1.26$ GPa are summarized in Table 1. In order to further substantiate our experimental findings, we performed a structural optimization within density functional theory and the generalized gradient approximation (GGA) of the exchange-correlation potential using the QUANTUM-ESPRESSO software package[34–36]. The Kohn-Sham orbitals were expanded in a plane-wave basis set with a kinetic-energy cutoff of 100 Ry. Spin-orbit coupling and a $U$ correction for the Ru $d$ orbitals of $U = 1.5$ eV have been taken into account. The Brillouin zone was sampled on a grid of $8 \times 8 \times 8$ $k$-points and integrated using the Marzari-Vanderbilt smearing with a smearing parameter of 0.02 Ry[37]. The empirical dispersion corrections DFT-D2 have been used to account for Van der Waals interactions. For the structural optimization, the lattice constants and the space group were kept fixed ($C2/m$ at ambient pressure and $R\bar{3}$ at 1.26 GPa), while the total energy and internal forces were optimized as a function of the allowed fractional coordinates. The $R\bar{3}$ space group arises from a defect-free rhombohedral stacking of the individual $p\bar{3}1m$ Cl-Ru-Cl layers. The results of these calculations are also included in Table 1, where they can be compared directly to the experimental values. We find excellent agreement between DFT and experiment, which confirms the experimentally determined structures and fractional coordinates.

To evaluate the effect of using different functionals in DFT, we performed the same calculations using plain GGA, GGA + SOC, and GGA + SOC + U, with and without the inclusion of DFT-D2 dispersion corrections. As shown in Supplementary Tables 4 and 5, the resulting structural parameters from all these calculations are in very good agreement, further supporting the validity of our results.

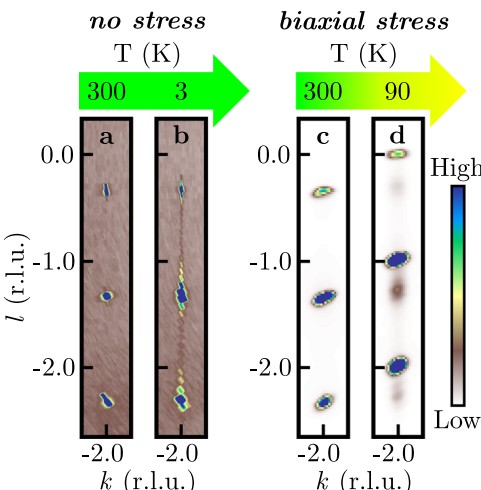

**Fig. 5 | Structural phase transition from monoclinic to rhombohedral symmetry driven by biaxial stress in bulk α-RuCl₃.** Reconstructed reciprocal space maps parallel to the 4−2$l$-plane (family 2) for the stress-free sample 1 (**a**),(**b**), and the sample 2 under biaxial stress (**c**), (**d**) are shown. Peaks at $l = n − 1/3$ and $l = n$ signal a cubic and a hexagonal closed chlorine packing, respectively.

The high-symmetry structure has been determined at RT, which naturally raises the question of its existence at low temperatures. To address this point, we conducted low-temperature experiments using two distinct sample mounting techniques. For sample 1, we placed it on a diamond anvil without any adhesive, specifically to avoid external stresses arising from differences in the thermal contraction between the sample and the holder during the cooling process. Conversely, sample 2 was securely affixed to an Al-holder. Consequently, at low temperatures, the different thermal contractions of α-RuCl₃ and Al generate a biaxial stress that aligns parallel to the $ab$-plane of sample 2.

As observed in the left panel of Fig. 5, the peak positions of the stress-free sample 1 remain unchanged upon cooling, indicating that this sample retains its monoclinic $C2/m$ phase. In contrast, sample 2 exhibits a distinct behavior: As shown in the right panel of Fig. 5, the reflections shift from $l = n − 1/3$ to integer $l$ values, very similar to what we have observed as a function of hydrostatic pressure. This shift in the $l$-direction is a clear indicator of the trigonal high-symmetry phase.

Based on the above findings, it seems very likely that biaxial stress indeed has the ability to trigger a structural transition from the $C2/m$ phase to the high-symmetry phase. The perspective that the latter can be readily attained at low temperatures in this way is obviously essential for the realization of a QSL. Furthermore, this makes the high-symmetry phase with $p\bar{3}1m$ layer symmetry accessible to a wide range of experimental techniques, hence, facilitating its thorough exploration.

On a more technical note, however, this also means that care must be taken when mounting α-RuCl₃ for low $T$ measurements. The strain dependence may, in fact, be the reason why some previous studies reported the occurrence of a first-order structural phase transition during cooling[38,39] or a rhombohedral structure[16,40]. It is remarkable that biaxial stress parallel to the RuCl₃-layers affects the stacking in the perpendicular direction. Up to our knowledge, the underlying physical mechanism is presently not known and certainly deserves further study.

## Discussion

We have developed a novel approach for the structure determination of transition metal trihalides, leveraging both the Bragg and diffuse scattering. Applying it to α-RuCl₃, we have uncovered an intriguing structural transition characterized by a change in the

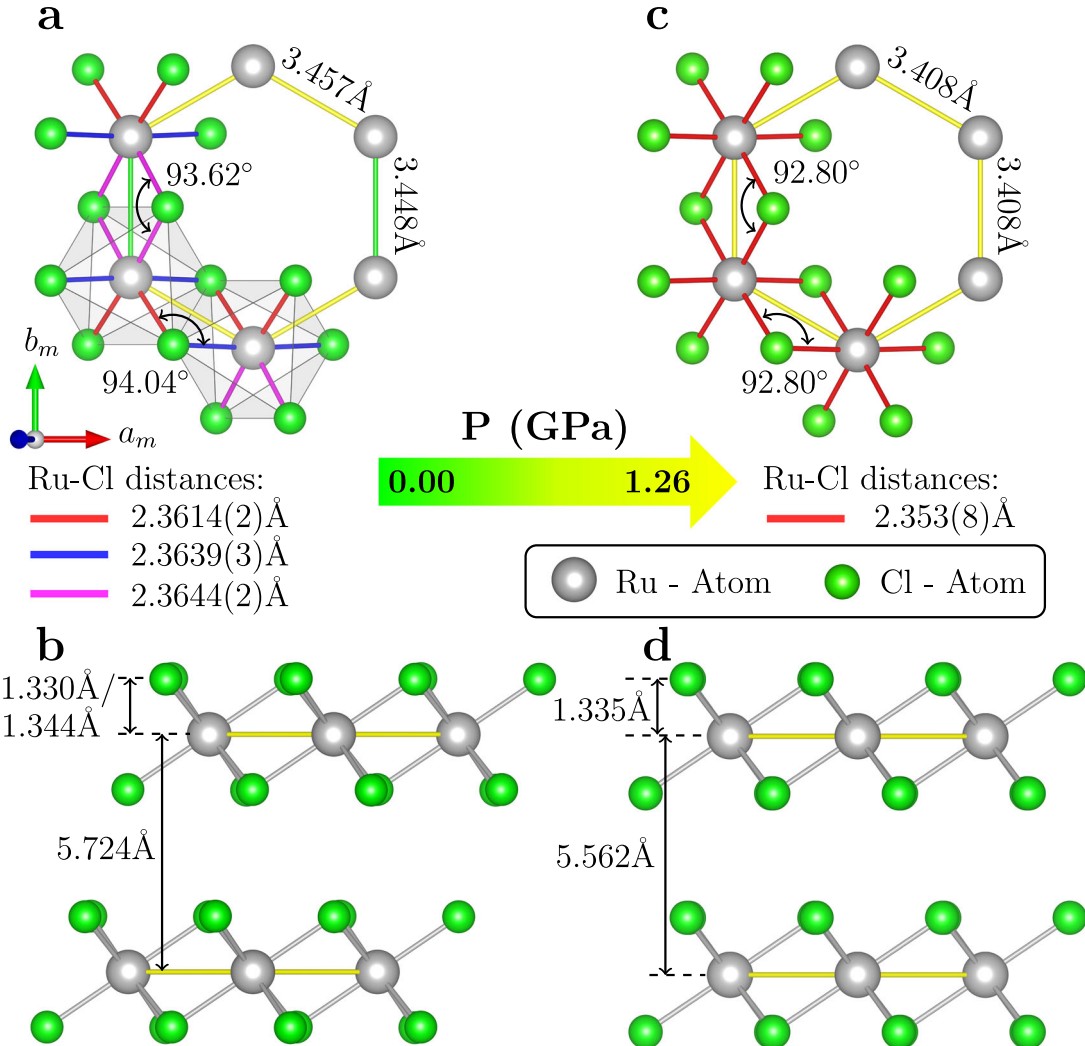

**Fig. 6 | The crystal structure of $\alpha$-RuCl$_3$ determined by analyzing the Bragg scattering as well as the diffuse scattering of high-quality single crystals at 0.0 GPa and 1.26 GPa.** The local Cl-Ru-Cl layer geometry is depicted in (**a**), (**c**), and two successive honeycomb layers viewed normal to the stacking direction are shown in (**b**), (**d**) for the monoclinic C2/m structure at 0.0 GPa and the high- symmetry structure with $p\bar{3}1m$ layer symmetry at 1.26 GPa, respectively. Equivalent nearest neighbor Ru-Ru links and Ru-Cl bond distances are encoded in the same colors in (**a**) and (**b**). For improved clarity, only the two nearest neighbor edge- sharing RuCl$_6$ octahedra are shown in (**a**).

stacking of the RuCl$_3$-layers. Notably, this transition involves huge parallel displacements of the RuCl$_3$-layers by approximately 2 Å along the ab-plane, while preserving the crystallinity of the RuCl$_3$-layers. We attribute this behavior to the very anisotropic chemical bonding in $\alpha$-RuCl$_3$ with strongly covalent RuCl$_3$-layers, which are weakly bonded along c. Similar transitions may, therefore, be expected to occur also in other layered compounds. Indeed, a comparable alteration of the stacking order has been observed in CrI$_3$[41], indicating that this phenomenon is not exclusive to $\alpha$-RuCl$_3$ but more prevalent among transition metal trihalides. In order to investigate these phenomena within a broader context, the experimental strategy developed here can serve as a powerful and versatile tool.

The pressure-induced structural transformation revealed in this study is illustrated in more detail in Fig. 6. The ambient pressure monoclinic *C2/m* structure exhibits distortions of the RuCl$_3$-layers, leading to variations in Ru–Cl distances, Ru–Ru distances, and Ru–Cl–Ru bond angles, see Fig. 6a, b. However, upon increasing the pressure to 1.26 GPa, these variations disappear. The RuCl$_3$-layers now adopt a trigonal structure with a single Ru–Cl distance, a single Ru–Ru distance, and a single Ru–Cl–Ru bond angle, as indicated in Fig. 6c, d. Consequently, the pressure-induced trigonal phase exhibits an

undistorted Ru-honeycomb lattice, characterized by a Ru−Cl−Ru bond angle of 92.80°, which is closer to the ideal value of 90°.

Based on the structural parameters for the high-symmetry phase determined above, an extensive theoretical analysis of the magnetic couplings has recently been conducted using state-of-the-art quantum chemistry calculations[42]. These numerical calculations reveal an exceptionally large $K/J$ ratio of 124, surpassing all previously reported values. The findings outlined in Ref. 42 underscore the highly favorable conditions for a QSL in the high-symmetry phase, characterized by nearly ideal $j_{\text{eff}} = 1/2$ states and an extremely small Heisenberg coupling $J$. Both of these factors play a crucial role in the emergence of a quantum spin liquid (QSL). Furthermore, the capability to stabilize the high-symmetry phase of $\alpha$-RuCl$_3$ through biaxial stress presents exciting opportunities for experimental exploration of this putative QSL using a diverse range of low-temperature techniques.

## Methods
### Crystal growth
$\alpha$-RuCl$_3$ single crystals were grown from phase-pure commercial $\alpha$- RuCl$_3$ powder via a high-temperature vapor transport technique[20]. We confirmed the monoclinic *C2/m* structure at ambient conditions for a

disorder-free crystal (crystal 1) by means of single crystal XRD (for details, see Supplementary Note 1). The obtained crystallographic parameters are fully consistent with previously published results[15].

## X-ray diffraction experiments

High-pressure XRD studies were carried out at beamlines ID15B and ID27 of the European Synchrotron Radiation Facility (ESRF) in Grenoble. To provide nearly hydrostatic pressure conditions, the membrane-driven diamond anvil cell (DAC) was loaded with helium as a pressure-transmitting medium. In the pressure range studied here and at RT, this provides essentially ideal hydrostatic conditions. The pressure inside the DAC was monitored in situ using the $R_{1,2}$ fluorescence of Cr-centers in ruby spheres placed next to the sample. The high-pressure XRD was done in transmission geometry with the DAC mounted on a single-axis goniometer with the rotation axis ($\omega$) perpendicular to the scattering plane. The diffracted radiation was recorded with a MAR555 flat panel detector at ID15B and a MAR-CCD detector at ID27 installed perpendicular to the primary beam.

Single crystal XRD data at room temperature were collected at ID15B, using monochromatic radiation of 30 keV ($\lambda = 0.4113$ Å) and a spot size of $10 \times 10$ $\mu m^2$ [red diamonds in Fig. 1b]. Diffraction data were recorded in steps of approximately 0.2 GPa up to 2 GPa. Each data set contains 120 frames with 0.5° scan width and an exposure time of 1 s per frame over a sample rotation of 60° ($-30° \leq \omega \leq 30°$). Using a very similar experimental setup at beamline ID27, the structural pressure-temperature phase diagram of $\alpha$-RuCl$_3$ was mapped out further at lower temperatures as well, using a continuous He-flow cryostat [red crosses in Fig. 1b]. In this experiment, the single crystalline sample was exposed to a monochromatic $3 \times 3$ $\mu m^2$ x-ray beam with a photon energy of 33 keV ($\lambda = 0.3738$ Å), while continuously recording the diffracted intensity on the detector during an $\omega$-movement of 60° ($-30° \leq \omega \leq 30°$).

## Density functional Theory

Density functional theory calculations were performed using the QUANTUM-ESPRESSO software package in version v7.3[34–36]. The generalized gradient approximation (GGA) of the exchange-correlation potential was used[34]. The Kohn-Sham orbitals were expanded in a plane-wave basis set with a kinetic-energy cutoff of 100 Ry. Spin-orbit coupling and a Hubbard $U$ correction for the Ru $d$ orbitals of $U = 1.5$ eV have been taken into account using the GGA + SOC and GGA + SOC + U extensions to the GGA. The Brillouin zone was sampled on a grid of $8 \times 8 \times 8$ $k$-points and integrated using the Marzari-Vanderbilt smearing method with a smearing parameter of 0.02 Ry[37]. The empirical dispersion corrections DFT-D2 have been used to account for Van der Waals interactions.

## Data availability

The data generated in this study are available in the main article and the Supplementary Information file. Source data are provided in this paper.

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

## Acknowledgements

We thank L. Janssen and M. Ruck for fruitful discussions. This research has been supported by the Deutsche Forschungsgemeinschaft through SFB 1143 (project-id 247310070), the Würzburg-Dresden Cluster of Excellence on Complexity and Topology in Quantum Matter-ct.qmat (EXC 2147, project-id 390858490) and SFB 1415 (project-id 417590517). We also gratefully acknowledge the support provided by the DRESDEN-concept alliance of research institutions and thank the ESRF for providing beamtime at ID27 and ID15B. For their support during the synchrotron experiments, we would like to thank M. L. Amigó, L. Leißner, and T. Raman.

## Author contributions

Q.S., T.R. and J.G. conceived the high-pressure diffraction experiment. A.I. and T.D. synthesized and provided $\alpha$-RuCl$_3$ single crystals. Q.S., T.R., G.G., F.C. and J.G. conducted the diffraction experiment at the ESRF. Q.S. analyzed the diffraction data and performed the structure determination. Q.S., T.R. and J.G. interpreted the obtained results. The manuscript has been written with contributions from all authors.

## Funding

## Competing interests

The authors declare no competing interests.
