## [Peer Review File · Nature Communications]

Pressure-tuning of α -RuCl₃ towards a quantum spin liquidREVIEWER COMMENTS

Reviewer #1 (Remarks to the Author):

Spin liquid state is a unique state - its propagating excitations possess only spin degrees of freedoms and the interacting quantum spins don't order even at zero temperature. Realization such a state in a real material is of importance because it is expected to have a potential application in topological quantum computing. This paper reports the high-pressure experimental results of the α - RuCl_3 material, one of the candidates of spin liquid materials, through the measurements of the Bragg and diffuse scattering of synchrotron radiation. Authors found that pressure induces a structural phase transition from the monoclinic $C2/m$ phase to a trigonal phase. These results are in good agreement with the results of the high-pressure heat capacity measurements at ~ 1 GPa where the AFM state is suppressed [Wang et al PRB 97 (2018) 245149] and the high-pressure Raman measurements at 1.1 GPa where a new peak has been observed [Li et al PRB 3 (2019) 023601]. Analysis of the data obtained at different pressures indicates that the Ru-Cl-Ru bond angle changes from 94° to 92.8° at 1.26 GPa. The pressure-induced the reduction of Ru-Cl-Ru bond angle gets close to the theoretically predicted value (90°) for an ideal Kitaev spin liquid state, suggesting that the approach used for tuning the structure toward to an actual spin liquid material is new. I would recommend to consider this work for publication.

Further comments:

I suggest authors to cite these two papers [[Wang et al PRB 97 (2018) 245149 and Li et al PRB 3 (2019) 023601] in their revised manuscript.

Reviewer #2 (Remarks to the Author):

In this work, Q. Stahl and co-workers report X-ray diffraction (XRD) and density functional theoretical (DFT) studies on α - RuCl_3 as a function of external pressure. α - RuCl_3 is one of the most eagerly pursued 'first-generation' proximate Kitaev candidate to realize the Kitaev

quantum spin liquid (QSL) state. Despite showing the thermodynamic and scattering signatures of spin fractionalization, the material ultimately orders magnetically at lower temperatures due to the presence of non-Kitaev (Heisenberg) interactions. Interestingly, external pressure has proven to be a potential perturbation to de-stabilize the ground state in α -RuCl₃ by tuning the exchange interactions by tuning the crystal structure. Like the other Kitaev materials, RuCl₃ also undergoes pressure induced dimerization of the honeycomb layer as established by several earlier reports, both experimental and theoretical.

In this paper, the authors have performed XRD studies on single crystals of α -RuCl₃ at varying pressures up to 2 GPa. By analyzing the Bragg and diffuse scattering spectra, they have shown that at room temperature the crystal undergoes a transformation from monoclinic C2/m structure with distorted honeycomb layers to an undistorted trigonal structure. To support their experimental claims, the authors have put forward brief DFT calculation under generalized-gradient approximation (GGA) for structural optimization. The authors have shown that application of biaxial stress can stabilize the high-symmetry phase at lower temperatures.

I think that the present results are interesting, but I have doubts and concerns regarding the presentation of the paper and consistency in analysis.

My questions and comments are as follows:

1) First of all, I felt that in the 'Introduction' section the authors did not render the motivation of their study strongly in the light of the existing high-pressure studies on the material under investigation. The previously reported high-pressure XRD, susceptibility, NMR, heat-capacity, Raman scattering, and optical reflectivity measurements as well as DFT calculations have already uncovered many crucial and critical aspects of the pressure evolution of the magnetic state of this material. A well-interpreted roadmap of the previous high-pressure reports was needed to emphasize the necessity and importance of the present work. When the authors are making a comment in the abstract like "This not only resolves contradictory findings in the literature", first they need to properly address what are the controversies in the literature and then how the present study adds to the puzzle. I did not find reasons why are the related works by Cui et al. [Phys. Rev. B 96, 205147 (2017)],

Biesner et al. [Phys. Rev. B 97, 220401(R) (2018)], Wang et al. [Phys. Rev. B 97, 245149 (2018)], and Li et al. [Phys. Rev. Mater. 3, 023601 (2019)] not cited in the present manuscript. Especially, Li et al. qualitatively proposed the high-pressure space group P-3m1 for RuCl₃ in their Raman spectroscopic measurements, so the authors should have referred to the same. Also, RuCl₃ is not the only candidate Kitaev material to reveal pressure induced increase in the proximity to the ideal Kitaev limit. Similar behavior is also observed in the recent high-pressure reports on the Kitaev candidate Cu₂IrO₃ in the stability range of its low-pressure monoclinic phase. The same should have been referred to the current manuscript.

2) In the caption of FIG. 1 the authors have mentioned that “The red crosses indicate points where overview scans have been collected (further explanations in the text)”. But I did not find an explicit detail on the same in the text.

3) The major flaw in the data interpretation I found was that only selected data has been chosen conveniently to be analyzed and shown. The authors have two more RT data points at pressures higher than 1.26 GPa as can be seen in FIG. 1 (b). But they did not put forward any comments or analysis on the same. Could the authors fit the data at the highest pressure to the already reported dimer phase? If so, it had to be explained in detail. And if not, then this study is in direct contradiction to the earlier findings, and it is the responsibility of the authors to explain the same.

4) While interpreting data in FIG. 2 (g)-(l) the authors have mentioned that “a notable broadening of the peaks is observed, accompanied by intensity spreading along the l-direction”. Intensity spreading along l-direction is clear, but what else broadening the authors are referring to? Is it along the k-direction? To my bare eyes, I could not detect any significant broadening for the intensity maxima unless the authors had demonstrated the same by showing the Intensity vs 2θ diffraction profiles.

5) For the analysis of the Bragg intensities, can the authors include the complete diffraction pattern along with the fitted profile from the refinement?

6) In the analysis of diffuse scattering, the authors have shown that “We find the best overall agreement between model and experiment for a Cl-displacement $\delta x_{Cl}=0.0102$ ”. Can the authors comment on why this particular value of δx_{Cl} gave the best match?

7) Another major drawback of the present study is the DFT calculation which is inaccurate and incomplete. The present DFT calculation is done under GGA only. The very basic realization behind the formation of $j_{eff} = 1/2$ Mott state in Kitaev spin liquid candidates includes moderately high spin-orbit coupling (SOC) and on-site Coulomb correlation (U_{eff}) in these materials. This is why all the DFT studies, not only on RuCl₃ but also in all

the other candidate Kitaev materials, (I do not feel the necessity to cite them here since there are numerous of them) include SOC and Hubbard-U parameters. The most concerning thing is that inclusion of SOC+U to GGA, drastically changes the electronic band structure of the material which directly affects the structural optimization. It seems to be utterly surprising how the DFT calculations presented in this study with GGA only gave such excellent agreement to the experimental lattice parameters. In order to be more reliable, the authors need to perform DFT calculations with GGA+SOC+U. Also, it is absolutely necessary that the authors show from DFT that the experimentally obtained phase transformation is energetically favored by calculating the pressure dependence of the enthalpies of the structures that they obtained from experiment.

8) In the details of the DFT calculation the authors have mentioned that “For the structural optimization, the lattice constants and the space group were kept fixed (C2/m at ambient pressure and R-3 at 1.26 GPa)”. Whereas everywhere else they mention the high-pressure space group as P-3m1. Is this discrepancy due to layer group and crystal space group? If so, the same needs to be explained explicitly.

9) In the study of the high-symmetry phase at low temperatures, there is no analysis and fitting details presented for the low-T high-symmetry structure (with biaxial stress). To make the statement “we demonstrate that this phase can also be stabilized by a slight biaxial pressure at low temperatures”, it is essential to demonstrate how the low-T high-symmetry structure (with biaxial stress) compares to the RT high-P high-symmetry structure.

10) For a clearer understanding of the effect of pressure, it is necessary to depict the pressure evolution of the lattice parameters, Ru-Ru distances, and Cl-Ru-Cl angles instead to just mentioning the values at two pressure values, ambient and 1.26 GPa. Also, the pressure evolution of the RuCl₆ octahedra needs to be investigated since the ideal O_h symmetry favors Kitaev exchange over Heisenberg exchange.

Though the present research has potential to contribute towards substantial advance in the respective field, I did not find the presentation, data analysis, and theoretical calculations adequate for the standards of the journal. In conclusion, I do not recommend publication of this work in Nature Communications.

Response Letter for manuscript NCOMMS-23-26428)

We sincerely thank the reviewers for their time and effort in reviewing our manuscript. We are pleased that both reviewers jointly acknowledge the general interest and potential impact of our manuscript. The reviewers raised several important points that were extremely helpful in improving our manuscript. We have carefully considered all the reviewers' points and revised the manuscript accordingly as explained in detail below. We hope that all reviewers will be satisfied with our revisions.

Point-by-Point response to Reviewer #1

(Comments by the Referees appear in green, our responses in black)

Spin liquid state is a unique state - its propagating excitations possess only spin degrees of freedoms and the interacting quantum spins don't order even at zero temperature. Realization such a state in a real material is of importance because it is expected to have a potential application in topological quantum computing. This paper reports the high-pressure experimental results of the α -RuCl₃ material, one of the candidates of spin liquid materials, through the measurements of the Bragg and diffuse scattering of synchrotron radiation. Authors found that pressure induces a structural phase transition from the monoclinic C2/m phase to a trigonal phase. These results are in good agreement with the results of the high-pressure heat capacity measurements at ~ 1 GPa where the AFM state is suppressed [Wang et al PRB 97 (2018) 245149] and the high-pressure Raman measurements at 1.1 GPa where a new peak has been observed [Li et al PRB 3 (2019) 023601]. Analysis of the data obtained at different pressures indicates that the Ru-Cl-Ru bond angle changes from 94° to 92.8° at 1.26 GPa. The pressure-induced the reduction of Ru-Cl-Ru bond angle gets close to the theoretically predicted value (90°) for an ideal Kitaev spin liquid state, suggesting that the approach used for tuning the structure toward to an actual spin liquid material is new. I would recommend to consider this work for publication.

We would like to thank reviewer #1 for her or his careful review of our manuscript and gratefully appreciate the highly positive assessment of our work. We are very happy to hear that he or she recommends our work for publication in Nature Communications.

Further comments:

I suggest authors to cite these two papers [[Wang et al PRB 97 (2018) 245149 and Li et al PRB 3 (2019) 023601] in their revised manuscript.

We thank the referee #1 for pointing us to these important papers, which we now cite appropriately in the revised version of our manuscript (see also our response to point 1 of referee #2)

Point-by-Point response to Reviewer #2

(Comments by the Referees appear in green, our responses in black)

In this work, Q. Stahl and co-workers report X-ray diffraction (XRD) and density functional theoretical (DFT) studies on α -RuCl₃ as a function of external pressure. α -RuCl₃ is one of the

most eagerly pursued 'first-generation' proximate Kitaev candidate to realize the Kitaev quantum spin liquid (QSL) state. Despite showing the thermodynamic and scattering signatures of spin fractionalization, the material ultimately orders magnetically at lower temperatures due to the presence of non-Kitaev (Heisenberg) interactions. Interestingly, external pressure has proven to be a potential perturbation to de-stabilize the ground state in α -RuCl₃ by tuning the exchange interactions by tuning the crystal structure. Like the other Kitaev materials, RuCl₃ also undergoes pressure induced dimerization of the honeycomb layer as established by several earlier reports, both experimental and theoretical.

In this paper, the authors have performed XRD studies on single crystals of α -RuCl₃ at varying pressures up to 2 GPa. By analyzing the Bragg and diffuse scattering spectra, they have shown that at room temperature the crystal undergoes a transformation from monoclinic C2/m structure with distorted honeycomb layers to an undistorted trigonal structure. To support their experimental claims, the authors have put forward brief DFT calculation under generalized-gradient approximation (GGA) for structural optimization. The authors have shown that application of biaxial stress can stabilize the high-symmetry phase at lower temperatures.

I think that the present results are interesting, but I have doubts and concerns regarding the presentation of the paper and consistency in analysis.

We would like to thank the reviewer #2 for the thorough review of our work. We are pleased that the reviewer assesses our results as interesting. However, he or she expresses a number of concerns, which we thoroughly address below:

My questions and comments are as follows:

1) First of all, I felt that in the 'Introduction' section the authors did not render the motivation of their study strongly in the light of the existing high-pressure studies on the material under investigation. The previously reported high-pressure XRD, susceptibility, NMR, heat-capacity, Raman scattering, and optical reflectivity measurements as well as DFT calculations have already uncovered many crucial and critical aspects of the pressure evolution of the magnetic state of this material. A well-interpreted roadmap of the previous high-pressure reports was needed to emphasize the necessity and importance of the present work. When the authors are making a comment in the abstract like "This not only resolves contradictory findings in the literature", first they need to properly address what are the controversies in the literature and then how the present study adds to the puzzle. I did not find reasons why are the related works by Cui et al. [Phys. Rev. B 96, 205147 (2017)], Biesner et al. [Phys. Rev. B 97, 220401(R) (2018)], Wang et al. [Phys. Rev. B 97, 245149 (2018)], and Li et al. [Phys. Rev. Mater. 3, 023601 (2019)] not cited in the present manuscript. Especially, Li et al. qualitatively proposed the high-pressure space group P-3m1 for RuCl₃ in their Raman spectroscopic measurements, so the authors should have referred to the same. Also, RuCl₃ is not the only candidate Kitaev material to reveal pressure induced increase in the proximity to the ideal Kitaev limit. Similar behavior is also observed in the recent high-pressure reports on the Kitaev candidate Cu₂IrO₃ in the stability range of its low-pressure monoclinic phase. The same should have been referred to the current manuscript.

We would like to thank the referee for these constructive comments. Indeed, the Introduction has fallen short on some of the previously reported studies on RuCl_3 . In the new version we have revised this part thoroughly and, in particular, discuss the important papers mentioned by the referee. It is correct that Li et al. suggests the spacegroup $P3_112$ for a phase that they find between 1.1 and 1.7 GPa at room temperature. This pressure range indeed corresponds approximately to the region where we find the high-symmetry phase as described in the main text. However, it is important to note that one key result of our study is that the layered structure of RuCl_3 is characterized by intrinsic stacking faults, which means that strictly speaking the translation symmetry along the out-of-plane direction is broken and one cannot clearly assign a 3D space group to the system. That is where the strength of our analysis of the diffuse scattering comes about: Using not only the scattered x-ray intensity in the sharp Bragg peaks but also the diffuse intensity in the streaks along the l -direction we were able to fully determine the structure of the Cl-Ru-Cl layers. It should be also clear that Raman measurements alone, as reported by Li et al., cannot provide such detailed structural information. Accordingly, our study answers directly the important question of the structure of the pressure-induced high-symmetry phase. In the new version of our introduction we make this point very clear.

Regarding the statement “This not only resolves controversies in the literature ...” in the abstract we would like to point out that the low temperature structure at ambient pressure of RuCl_3 is indeed still being intensively discussed with at least three suggested spacegroups ($R\bar{3}$, $C2/m$ and $P3_112$) [1–5]. Therefore, our finding that even subtle biaxial strain trigger a structural transition might provide a natural explanation of why different low-temperature phases have been reported in the literature. We have rephrased this sentences to emphasize that we are addressing the controversies regarding the low-temperature structure of RuCl_3 .

2) In the caption of FIG. 1 the authors have mentioned that “The red crosses indicate points where overview scans have been collected (further explanations in the text)”. But I did not find an explicit detail on the same in the text.

Thank you for pointing out this obvious inconsistency. The overview scans represent a single frame continuously recorded while the sample was rotated about 60° . The corresponding explanation is indeed found in the “methods” section rather than in the main text. We have corrected this in the revised version.

3) The major flaw in the data interpretation I found was that only selected data has been chosen conveniently to be analyzed and shown. The authors have two more RT data points at pressures higher than 1.26 GPa as can be seen in FIG. 1 (b). But they did not put forward any comments or analysis on the same. Could the authors fit the data at the highest pressure to the already reported dimer phase? If so, it had to be explained in detail. And if not, then this study is in direct contradiction to the earlier findings, and it is the responsibility of the authors to explain the same.

While we do indeed focus on the pressure induced high-symmetry, we have by no means “chosen conveniently” data to be analyzed. This is absolutely not the case. As explained in more detail in our reply to point 10 it turns out that only the data at near ambient pressure and

at $p = 1.26$ GPa can be analyzed using our method. Furthermore, it is correct that we have measured one additional data point at 1.5 GPa. From the supplementary Fig. 8 it is very clear that the sample has transformed into the dimerized phase at this pressure. This phase has been reported and carefully characterized previously (Ref. 22). To address the referee's concern we now discuss the dimerization in the main text of the revised manuscript.

4) While interpreting data in FIG. 2 (g)-(l) the authors have mentioned that “a notable broadening of the peaks is observed, accompanied by intensity spreading along the l -direction”. Intensity spreading along l -direction is clear, but what else broadening the authors are referring to? Is it along the k -direction? To my bare eyes, I could not detect any significant broadening for the intensity maxima unless the authors had demonstrated the same by showing the Intensity vs 2Θ diffraction profiles.

We thank the Referee for pointing out this unfavorably worded sentence. Indeed, the diffuse stripes were only observed along the l -direction, indicating a lack of long-range order of the stacking. As the pressure increases the peaks become broader along the l -direction and the center of mass of the peak shifts from $l = n + 1/3$ to $l = n$. We have changed the corresponding sentence from “accompanied by intensity spreading along the l -direction” to “accompanied by a shift of the intensity maximum along the l -direction”.

5) For the analysis of the Bragg intensities, can the authors include the complete diffraction pattern along with the fitted profile from the refinement?

As far as we understand the referee he or she is asking for a visualization of the recorded XRD intensity along with the simulated one based on our structure model. This is very hard to achieve as we are dealing with full single crystal data sets in 3D k -space as opposed to powder diffraction data where one could easily judge the quality of the fit from a comparison between the 1D experimental diffraction profile (intensity versus scattering angle) and the theoretical one. In full agreement with the standards of single crystal x-ray crystallography we summarize the common statistical quantities to assess the quality of the refinement like R_{int} value, GOF and so on in Table II and III of the supplementary information. Additionally, we provide the commonly used F_{obs} versus F_{cal} plot in Fig. 1.

6) In the analysis of diffuse scattering, the authors have shown that “We find the best overall agreement between model and experiment for a Cl-displacement $\delta x_{Cl} = 0.0102$ ”. Can the authors comment on why this particular value of δx_{Cl} gave the best match?

As far as we can see there are two ways to read this question: On the one hand the question could be aimed at why the diffuse scattering is sensitive to the parameter δx_{Cl} such that it is a suitable means of measuring this parameter. The structure factor for a single Cl-Ru-Cl layer ($p\bar{3}1m$ layer group) obviously depends on the parameter δx_{Cl} ($F \propto \exp(h \cdot \delta x_{Cl})$). As outlined in the supplementary information the diffuse scattering is then simulated for a stack of 1000 layers which incorporates the stacking faults. As shown in Fig. 4 of the main text the interference from these layers leads to a diffuse x-ray intensity along l -direction which indeed is very sensitive to the δx_{Cl} parameter.

On the other hand the question could be aimed at how we chose the best fitting model:

Figure 1: Theoretical squared structure factors $|F_{cal}|^2$ versus observed $|F_{obs}|^2$ squared structure factors for the structural model determined at $p = 1.26$ GPa taking only the sharp Bragg peaks into account.

The diffuse scattering streaks along the l -direction have been simulated for different discrete values of δx_{Cl} in steps of 3 pm. The best match has then been chosen upon manual visual comparison with the experimental data and an essentially perfect fit for $\delta x_{Cl} = 0.0102$ has been found as can be seen from Fig. 4 of the main text. A full χ^2 -optimization is not necessary and also not feasible because the algorithm to simulate the theoretical intensity profiles is numerically demanding and cannot easily be called frequently. However, the approach of visually comparing a simulation to experimental data is not uncommon, rather it is often used to analyze for instance x-ray absorption spectroscopy data or angle-resolved photo electron emission spectroscopy data. Therefore, we are confident in the validity of the conclusion that $\delta C_{Cl} = 0.0102$ provides the best theoretical description of the observed data among the δx_{Cl} values for which simulations were performed.

7) Another major drawback of the present study is the DFT calculation which is inaccurate and incomplete. The present DFT calculation is done under GGA only. The very basic realization behind the formation of $j_{eff} = 1/2$ Mott state in Kitaev spin liquid candidates includes moderately high spin-orbit coupling (SOC) and on-site Coulomb correlation (U_{eff}) in these materials. This is why all the DFT studies, not only on $RuCl_3$ but also in all the other candidate Kitaev materials, (I do not feel the necessity to cite them here since there are numerous of them) include SOC and Hubbard- U parameters. The most concerning thing is that inclusion of SOC+ U to GGA, drastically changes the electronic band structure of the material which directly affects the structural optimization. It seems to be utterly surprising how the DFT calculations presented in this study with GGA only gave such excellent agreement to the

experimental lattice parameters. In order to be more reliable, the authors need to perform DFT calculations with GGA+SOC+U. Also, it is absolutely necessary that the authors show from DFT that the experimentally obtained phase transformation is energetically favored by calculating the pressure dependence of the enthalpies of the structures that they obtained from experiment.

Indeed, both spin-orbit coupling (SOC) and electron-electron correlations are very important in describing the electronic structure and magnetic properties of RuCl_3 correctly. On the one hand, GGA+SOC can reasonably well account for SOC by using a fully relativistic version of the Kohn-Sham equations. The calculations presented in our manuscript were actually done within this GGA+SOC scheme which was indeed not clearly stated in the previous version of the manuscript. On the other hand, electron-electron interactions are only crudely approximated by GGA(+SOC)+U. We would also like to emphasize that the presented DFT results were merely intended to support our experimentally found structure which represents the key information of our study. In other words the central results of our study comprise the experimentally found crystal structure and the non-standard method of diffuse diffraction analysis used to determine it. Both results are valid independently of the presented DFT simulations, which should only serve as a verification of our analysis.

However, the referee's remark prompted us to thoroughly analyze the effects of the different DFT approximation levels on the symmetry-constrained structural relaxations. Accordingly, we compared the structures obtained from plain GGA, fully relativistic GGA+SOC and fully relativistic GGA with on-site Hubbard correction GGA+SOC+U. Additionally, we also studied the effect of empirical DFT-D2 van der Waals dispersion-corrections as implemented in the quantum-espresso package [6, 7]. We used $U = 1.5 \text{ eV}$ for the U parameter as in previous DFT studies on this material [8]. To keep the numerical effort reasonable we sampled the Brillouin zone on mesh a of $8 \times 8 \times 8$ k -points and used the Marzari-Vanderbilt-DeVita-Payne smearing with a smearing parameter of 0.02 Ry. This aspect is different from the calculations presented in our initial manuscript. These were carried out on a 24×24 k -mesh using the optimized tetrahedron integration method.

A thorough comparison of the obtained results is shown in tables Tab. 1 and Tab. 2 and also incorporated into the revised version of the manuscript. For convenience the tables also contain the results from the previous calculations on the finer k -mesh in parentheses in the GGA+SOC column. As can be clearly seen the different DFT approximation levels (GGA, GGA+SOC, GGA+SOC+U) yield very similar optimized structures with the largest differences occurring from plain GGA to GGA+SOC. In particular, the crucial Cl-Ru-Cl binding angle varies only within ≈ 0.3 as a function of the Hubbard U and the DFT-D2 dispersion corrections. Overall, the optimized DFT(+SOC+U) structures are essentially equal within the precision of the method and very close to the experimentally found structure. We have also verified the GGA+SOC+U result for the R3b high-symmetry phase with the full potential FPLO DFT-code [9, 10] (see Fig. 2).

Accordingly, we are now even more convinced that our experimentally determined structures are well-supported by our DFT calculations.

In the revised version of our manuscript we clearly state that the DFT simulations are by no means central for our conclusion and should be merely understood as an additional support for our experimentally determined crystal structure. We also included the comparison of the different DFT approximation levels in the revised supplementary material.

Figure 2: Total Energy calculated as a function of δx_{Cl} for the R3b structure using the full potential DFT code FPLO. The fully relativistic GGA potential including spin orbit coupling and Hubbard U corrections were used.

8) In the details of the DFT calculation the authors have mentioned that “For the structural optimization, the lattice constants and the space group were kept fixed (C2/m at ambient pressure and R-3 at 1.26 GPa)”. Whereas everywhere else they mention the high-pressure space group as P-3m1. Is this discrepancy due to layer group and crystal space group? If so, the same needs to be explained explicitly.

We thank the referee for pointing out this seeming inconsistency. As correctly suspected by the referee, we can experimentally determined only the layer group of $RuCl_3$ to be P-3m1. This is because only the chlorine sublattices assume a well ordered stacking in the out-of-plane direction while the Ruthenium sublattices have a disordered stacking along the out-of-plane direction while still maintaining the well-defined layer symmetry P-3m1. This is in detail described in our manuscript and indeed is one of the central findings of our study.

Since we need a translationally invariant structure for our DFT simulations we choose a periodic approximation of the real disordered stacking structure. A rhombohedral stacking of the 2D P-3m1 layers yielding a 3D R3b space group is the best approximation of the disordered stacking. In the revised version of the manuscript we better explain Why the DFT calculations are done in the R3b space group.

9) In the study of the high-symmetry phase at low temperatures, there is no analysis and fitting details presented for the low-T high-symmetry structure (with biaxial stress). To make the statement “we demonstrate that this phase can also be stabilized by a slight biaxial pressure at low temperatures”, it is essential to demonstrate how the low-T high-symmetry structure (with biaxial stress) compares to the RT high-P high-symmetry structure.

	DFT		GGA+SO		GGA+SO+U (U=1.5 eV)	
	no vdW corr	DFT-D2	no vdW corr	DFT-D2	no vdW corr	DFT-D2
Ru						
x	0	0	0 (0)	0	0	0
y	0.16643	0.16651	0.166327 (0.166351)	0.166458	0.166497	0.166499
z	1/2	1/2	1/2 (1/2)	1/2	1/2	1/2
Cl ₁						
x	0.22099	0.22282	0.22618 (0.22692)	0.2279	0.22686	0.22841
y	0	0	0 (0)	0	0	0
z	0.72955	0.73055	0.73482 (0.73488)	0.73573	0.73515	0.73644
Cl ₂						
x	0.24985	0.25020	0.25005 (0.25059)	0.2504	0.2504	0.25032
y	0.17636	0.17558	0.17466 (0.17407)	0.17385	0.17416	0.17367
z	0.27259	0.27134	0.26788 (0.26709)	0.26667	0.26738	0.26594
Ru-Ru (Å)	3.44646	3.44774	3.44394 (3.44441)	3.44664	3.44745	3.44749
	3.45765	3.45701	3.45891 (3.45872)	3.45756	3.45716	3.45714
Ru-Cl-Ru (°)	94.9874	94.6555	93.6332 (93.5531)	93.3582	93.5693	93.2217
	95.4547	95.0348	94.2565 (94.0842)	93.8374	94.0206	93.6698

Table 1: Optimized atomic positions for the monoclinic phase (C2/m) for different levels of DFT approximation. Atomic coordinates refer to the conventional monoclinic cell with lattice parameters $a_m = 5.9875 \text{ \AA}$, $b_m = 10.3529 \text{ \AA}$, $c_m = 6.0456 \text{ \AA}$ and $\beta = 108.777$. The column highlighted in red corresponds to the GGA+SOC calculations that led to the structural parameters given in the original version of the manuscript. Note that in the earlier calculations we used a $24 \times 24 \times 24$ k -mesh and the optimized tetrahedron method for Brillouin zone integration (values in parentheses), while the present calculations use a $8 \times 8 \times 8$ k -mesh and the Marzari-Vanderbilt smearing for integration. This is the reason for the slightly different values.

	DFT		GGA+SO		GGA+SO+U (U=1.5 eV)	
	no vdW corr	DFT-D2	no vdW corr	DFT-D2	no vdW corr	DFT-D2
Ru						
x	2/3	2/3	2/3 (2/3)	2/3	2/3	2/3
y	1/3	1/3	1/3 (1/3)	1/3	1/3	1/3
z	0.49989	0.49992	0.50004 (0.50002)	0.49989	0.49995	0.49975
Cl ₁						
x	0.34738	0.34559	0.34351 (0.34291)	0.34181	0.34341	0.34175
y	0.00001	0.00008	0.00022 (0.00022)	0.00039	0.00018	0.00029
z	0.73657	0.73725	0.74056 (0.74045)	0.74111	0.74128	0.74178
Ru-Ru (Å)	3.40799	3.40799	3.40797 (3.40799)	3.40799	3.40799	3.40799
Ru-Cl-Ru (°)	93.7955	93.4768	92.7005 (92.6225)	92.4218	92.5696	92.3019

Table 2: Optimized atomic positions for the high symmetry phase (R3b) for different levels of DFT approximation. Atomic coordinates are mapped to the hexagonal unit cell of the layer space group P-3m1 ($a_h = b_h = 5.9028 \text{ \AA}$, $c_h = 5.562 \text{ \AA}$ and $\alpha = \beta = 90$, $\gamma = 120$). The column highlighted in red corresponds to the GGA+SOC calculations that led to the structural parameters given in the original version of the manuscript. Note that in the earlier calculations we used a $24 \times 24 \times 24$ k -mesh and the optimized tetrahedron method for Brillouin zone integration (values in parentheses), while the present calculations use a $8 \times 8 \times 8$ k -mesh and the Marzari-Vanderbilt smearing for integration. This is the reason for the slightly different values.

In order to investigate the existence of the high-symmetry phase at low temperature we have designed a special Al-holder that creates biaxial stress due to its thermal contraction during the cooling process. Unfortunately, this sample holder limits the accessible region in reciprocal space. Therefore, it is not possible to perform the same non-standard analysis of Bragg- and diffuse scattering as for the hydro static pressure data. However, for the 4-2 l peaks we observe the same behavior as we did as a function of hydro static pressure. Namely, a shift of the peaks from $l = n - 1/3$ to $l = n$ indicating the same transition of the chlorine stacking from cubic close packing to a hexagonal close packing. Since this is a very clear indicator for the transition into the high-symmetry phase we assume that it indeed occurs at low-temperature when the sample is subject to subtle biaxial stress.

However, we agree with the referee that this conclusion is not fully supported by our data. Accordingly, we have rephrased the corresponding statement in the revised version of the manuscript. The complete analysis of this biaxial pressure induced phase should certainly be scrutinized in follow-up studies.

10) For a clearer understanding of the effect of pressure, it is necessary to depict the pressure evolution of the lattice parameters, Ru-Ru distances, and Cl-Ru-Cl angles instead to just mentioning the values at two pressure values, ambient and 1.26 GPa. Also, the pressure evolution of the RuCl₆ octahedra needs to be investigated since the ideal O_h symmetry favors Kitaev exchange over Heisenberg exchange.

We agree with the referee that it would be desirable to have these parameters as a function

of pressure. However, as explained in the paper in the region between 0.2 GPa and 1.26 GPa the stacking of both, Ruthenium layers and Chlorine layers is disordered. This unfortunately, prevents to apply the analysis we have performed at $P = 1.26$ GPa.

List of changes (manuscript NCOMMS-23-26428)

1. Abstract: In the abstract we now specifically say that reported contradictory concerns the low temperature crystal structure of RuCl_3 .
2. ll. 53-55: In the introduction we added a sentence describing that the low-temperature crystal structure is not yet established and added corresponding references (Refs. 16-20 in the main text).
3. ll. 67-86: In the introduction we extended the discussion of previous pressure-dependent studies on RuCl_3 and relate them more directly to our study. We also added corresponding references (Refs. 23,24,26 in the main text).
4. ll. 148-152: In the section “experimental results” we corrected the sentence “. . . a notable broadening of the peaks is observed, accompanied by intensity spreading along the l -direction.” to “. . . a notable broadening of the peaks along l is observed, accompanied by a shift of the intensity maximum along the l -direction.”.
5. ll. 193-197: At the end of the “experimental results” section we added a paragraph describing the transformation into the dimerized phase at $p = 1.5$ GPa. We also added Fig. S2 to illustrate this transformation.
6. ll. 300-306: In the section “comparison to density functional theory” we updated the details of calculations as we now present the results for the GGA+SOC+U calculation including DFT-D2 dispersion corrections.
7. in Table I, the structural parameters have been updated to the results from the new GGA+SOC+U calculations.
8. ll. 318-325: In the section “comparison to density functional theory” we added a paragraph corresponding to the thorough comparison of different DFT approximation levels which we did for the revised manuscript version. Supplementary Tables S4 and S5 were added to summarize the results from our DFT survey.
9. ll. 345.-353: In the section “high-symmetry phase at low temperature” we rephrased the sentences describing the observed intensity shift of the reflections along l to emphasize that this is a key indicator for the formation of the high symmetry phase.
10. Ref. 43 has been published in PRB in the meantime. The corresponding information has been updated.
11. Supplementary Figure S1 has been added to illustrate the observed transition into the dimerized phase at $p = 1.5$ GPa.
12. Many minor corrections of typos and wording throughout the manuscript.

References

- [1] S.-Y. Park, S.-H. Do, K.-Y. Choi, D. Jang, T.-H. Jang, J. Scheffer, C.-M. Wu, J. S. Gardner, J. M. S. Park, J.-H. Park, et al., *Journal of Physics: Condensed Matter* **36**, 215803 (2024), ISSN 1361-648X.
- [2] Z. Wang, S. Reschke, D. Hübner, S.-H. Do, K.-Y. Choi, M. Gensch, U. Nagel, T. Rößler, and A. Loidl, *Physical Review Letters* **119**, 227202 (2017), ISSN 1079-7114.
- [3] M. He, X. Wang, L. Wang, F. Hardy, T. Wolf, P. Adelman, T. Brückel, Y. Su, and C. Meingast, *Journal of Physics: Condensed Matter* **30**, 385702 (2018), ISSN 1361-648X.
- [4] K. Ran, J. Wang, W. Wang, Z.-Y. Dong, X. Ren, S. Bao, S. Li, Z. Ma, Y. Gan, Y. Zhang, et al., *Physical Review Letters* **118**, 107203 (2017), ISSN 1079-7114.
- [5] A. Glamazda, P. Lemmens, S.-H. Do, Y. S. Kwon, and K.-Y. Choi, *Physical Review B* **95**, 174429 (2017), ISSN 2469-9969.
- [6] V. Barone, M. Casarin, D. Forrer, M. Pavone, M. Sami, and A. Vittadini, *Journal of Computational Chemistry* **30**, 934 (2009), <https://onlinelibrary.wiley.com/doi/pdf/10.1002/jcc.21112>, URL <https://onlinelibrary.wiley.com/doi/abs/10.1002/jcc.21112>.
- [7] S. Grimme, *Journal of Computational Chemistry* **27**, 1787 (2006), <https://onlinelibrary.wiley.com/doi/pdf/10.1002/jcc.20495>, URL <https://onlinelibrary.wiley.com/doi/abs/10.1002/jcc.20495>.
- [8] S. Biswas, Y. Li, S. M. Winter, J. Knolle, and R. Valentí, *Phys. Rev. Lett.* **123**, 237201 (2019), URL <https://link.aps.org/doi/10.1103/PhysRevLett.123.237201>.
- [9] K. Koepnick and H. Eschrig, *Phys. Rev. B* **59**, 1743 (1999).
- [10] H. Eschrig, M. Richter, and I. Opahle, *Relativistic Solid State Calculations, in: Relativistic Electronic Structure Theory, Part 2. Applications*, vol. 13 (Elsevier, 2004).

REVIEWERS' COMMENTS

Reviewer #1 (Remarks to the Author):

In the revised manuscript, the authors have included the suggested works (Refs. 24 and 26), but they have not provided the progress made in these studies. To enhance the manuscript's clarity, I suggest the authors to consider incorporating the results of pressure versus Neel temperature, as reported in Ref. 24, into Fig. 1b. This would be valuable in illustrating the transitional behavior of the sample from an ambient-pressure antiferromagnetic (AFM) phase to a quantum spin liquid state under pressure.

Reviewer #2 (Remarks to the Author):

The authors have done a satisfactorily wise job in addressing the concerns of the Referees. I am satisfied with their response to my comments. I would, therefore, recommend the publication of this revised manuscript in Nature Communications.

Only one mild point that I wanted to bring into the authors' note is the compilation error for the references in the supplementary information.

Response Letter for manuscript NCOMMS-23-26428A)

Point-by-Point response to Reviewer #1

(Comments by the Referees appear in green, our responses in black)

In the revised manuscript, the authors have included the suggested works (Refs. 24 and 26), but they have not provided the progress made in these studies. To enhance the manuscript's clarity, I suggest the authors to consider incorporating the results of pressure versus Neel temperature, as reported in Ref. 24, into Fig. 1b. This would be valuable in illustrating the transitional behavior of the sample from an ambient-pressure antiferromagnetic (AFM) phase to a quantum spin liquid state under pressure.

We would like thank Referee #1 once more for her or his careful review of our manuscript and the valuable comments. In the final version of our manuscript we added a sentence to describe the advances made in Refs. 24 and 26. Although we understand that it might be helpful to include the results from Ref. 24 in Fig. 1b to illustrate the relationship between magnetic properties and lattice structure, we prefer to keep the figure simple and show only the data from our study.

Point-by-Point response to Reviewer #2

(Comments by the Referees appear in green, our responses in black)

The authors have done a satisfactorily wise job in addressing the concerns of the Referees. I am satisfied with their response to my comments. I would, therefore, recommend the publication of this revised manuscript in Nature Communications. Only one mild point that I wanted to bring into the authors' note is the compilation error for the references in the supplementary information.

We are very happy to hear that reviewer #2 is satisfied by the revisions we made in the previous round. We would like to thank him or her again for the constructive input and the thorough review of our work.